# *Lactobacillus kefiranofaciens*: From Isolation and Taxonomy to Probiotic Properties and Applications

**DOI:** 10.3390/microorganisms9102158

**Published:** 2021-10-16

**Authors:** Marina Georgalaki, Georgia Zoumpopoulou, Rania Anastasiou, Maria Kazou, Effie Tsakalidou

**Affiliations:** Laboratory of Dairy Research, Department of Food Science and Human Nutrition, Agricultural University of Athens, Iera Odos 75, 118 55 Athens, Greece; gz@aua.gr (G.Z.); ranastasiou@aua.gr (R.A.); kmaria@aua.gr (M.K.); et@aua.gr (E.T.)

**Keywords:** *Lactobacillus kefiranofaciens*, subspecies discrimination, kefir, grains, kefiran, probiotic properties, growth media

## Abstract

One of the main lactic acid bacterial species found in the kefir grain ecosystem worldwide is *Lactobacillus kefiranofaciens*, exhibiting strong auto-aggregation capacity and, therefore, being involved in the mechanism of grain formation. Its occurrence and dominance in kefir grains of various types of milk and geographical origins have been verified by culture-dependent and independent approaches using multiple growth media and regions of the 16S rRNA gene, respectively, highlighting the importance of their combination for its taxonomic identification. *L. kefiranofaciens* comprises two subspecies, namely *kefiranofaciens* and *kefirgranum*, but only the first one is responsible for the production of kefiran, the water-soluble polysaccharide, which is a basic component of the kefir grain and famous for its technological as well as health-promoting properties. *L. kefiranofaciens*, although very demanding concerning its growth conditions, can be involved in mechanisms affecting intestinal health, immunomodulation, control of blood lipid levels, hypertension, antimicrobial action, and protection against diabetes and tumors. These valuable bio-functional properties place it among the most exquisite candidates for probiotic use as a starter culture in the production of health-beneficial dairy foods, such as the kefir beverage.

## 1. Introduction

Kefir is a viscous, slightly carbonated dairy beverage, which has its origins in the Caucasian, Tibetan and Mongolian mountains. It comprises a complex microbial consortium of mainly lactic acid bacteria (LAB), acetic acid bacteria (AAB) and yeasts, and is considered a functional dairy product as it has been associated with a wide range of health benefits [1]. Three *Lactobacillus* species have been identified in the microbiota of the traditional dairy product kefir, namely *Lentilactobacillus kefiri* (basonym: *Lactobacillus kefir*) [2], *Lactobacillus*
*kefiranofaciens* [3] and *Lentilactobacillus parakefiri* (basonym: *Lactobacillus parakefir*) [4].

*Lactobacillus* species are part of the microbiota of humans and animals, are found in a variety of food products and have been studied extensively as fermentation starter and/or adjunct cultures and probiotics. They are generally recognized as safe (GRAS) by the US Food and Drug Authority (FDA) and belong to the qualified presumption of safety (QPS) list of the European Food Safety Authority (EFSA) [5,6]. However, several species behave as opportunistic pathogens and have been involved in human infection cases [7].

Until March 2020, *Lactobacillus* was the largest and most diverse genus within LAB, accounting for 261 species. However, a polyphasic approach based on various criteria, such as core genome phylogeny, pairwise average amino acid identity, clade-specific signature genes, physiological criteria and ecology, was used recently for the re-assessment of the taxonomy of the families *Lactobacillaceae* and *Leuconostocaceae*, resulting in the reclassification of the genus *Lactobacillus* into 25 genera including the emended genus *Lactobacillus* [8]. *Lactobacillus* currently comprises 51 species [8], among them the species *Lactobacillus helveticus*, *L. kefiranofaciens*, *Lactobacillus delbrueckii* and *L. kefiri*, which are commonly found in fermented milks, such as kefir, koumiss and buttermilk [9].

The nomenclature of the *L. kefiranofaciens* species in the emended genus *Lactobacillus* remained unchanged and it is still taxonomically assigned to the genus *Lactobacillus*. *L. kefiranofaciens* comprises two subspecies and has been isolated not only from kefir grains but from other fermented dairy products as well, such as koumiss, hurunge and tarag. *L. kefiranofaciens* subsp. *kefiranofaciens*, with a genome size of 2.26 Mbp and mol% G + C content of DNA 37.2 for the type strain LMG 19149^T^, is the polysaccharide kefiran-producing subspecies and is a prominent member of the kefir microbiota [3,8,10]. *L. kefiranofaciens* subsp. *kefirgranum*, with a genome size of 2.10 Mbp and mol% G + C content of DNA 37.5 for the type strain LMG 15132^T^, is part of the core microbiota of kefir grains [4,8,10]. Strains of both subspecies are Gram-positive, non-motile, capsulated, non-spore-forming rods (generally 0.8 to 1.2 μm by 3.0 to 20.0 μm) that occur as single cells, in pairs, or occasionally in short chains [3].

This review discusses the taxonomic history of *L. kefiranofaciens*, complex issues related to its isolation and growth conditions, biochemical and physiological characteristics, ecological niches, and finally, current research and novel applications relevant to human health.

## 2. A Brief Isolation and Taxonomic History

In 1967, Rivière et al. were the first to describe a capsule-forming heterofermentative *Lactobacillus* species from Russian kefir grains purchased in Moscow in 1966 [11]. Based on biochemical and physiological characteristics, notably gas production from glucose, they classified this long rod-forming and polysaccharide-producing microorganism as *Lactobacillus brevis* and called its polysaccharide kefiran. Similarly, in 1986, Toba et al. managed to isolate a capsule-forming homofermentative *Lactobacillus* unclassified species from kefir grains obtained from Chr. Hansen’s Laboratory (Copenhagen, Denmark) using a synthetic growth medium [12]. *Lactobacillus*
*kefiranofaciens* sp. nov. was subsequently described by members of the same team in Tokyo, Japan [3]. Fujisawa et al. [3] obtained and analyzed four strains of the species, namely WT-2B, WT-6A, WT-7 and WT-8, isolated from kefir grains by Toba et al. [12]. They described *L. kefiranofaciens* sp. nov., with WT-2B (ATCC 43761) being assigned as the type strain. The isolated strains were characterized as Gram-positive, non-motile, capsulated, non-spore-forming, slime-forming, homofermentative, facultatively anaerobic and rod-shaped LAB, which differed from all the validly described homofermentative species of the genus *Lactobacillus* in the carbohydrate fermentation pattern [3].

In 1994, two new species, namely *Lactobacillus kefirgranum* sp. nov. and *Lactobacillus parakefir* sp. nov., were isolated in Japan from kefir grains obtained from Chr. Hansen’s Laboratory in Copenhagen, Denmark [4]. Both species were characterized as facultatively anaerobic, Gram-positive, non-motile, non-spore-forming rods occurring as single cells, in pairs, or occasionally in short chains. *L. kefirgranum* produced DL-lactic acid homofermentatively, while *L. parakefir* produced L-lactic acid and CO_2_ via heterofermentation, while hydrogen sulfide, catalase and oxidase were not produced by any of the species. In 2004, 14 homofermentative LAB were isolated from kefir grains and kefir fermented milk samples and were assigned to either *Lactobacillus*
*kefiranofaciens* or *Lactobacillus kefirgranum*, based on their characteristic morphotypes, phenotypic features and sodium dodecyl sulfate-polyacrylamide gel electrophoresis (SDS-PAGE) profiles of whole-cell proteins [10]. Vancanneyt et al. reported that *L. kefiranofaciens* and *L. kefirgranum* shared 100% 16S rRNA gene sequence similarity and they both belonged phylogenetically to the *Lactobacillus acidophilus* species group, along with *Lactobacillus crispatus*, *Lactobacillus gallinarum*, *Lactobacillus hamsteri*, *Lactobacillus amylovorus*, *Lactobacillus amylolyticus*, *Lactobacillus intestinalis* and *L. acidophilus*, sharing a similarity of 95.6–97.7% [10]. Moreover, their DNA–DNA binding values of >79% and analogous DNA G + C content of 37–38 mol% showed that the isolates belonged to the species *L. kefirgranum*, which is a later synonym of *L. kefiranofaciens*. Thereafter, an emended description was proposed for the species *L. kefiranofaciens*, while two novel subspecies were described, *L. kefiranofaciens* subsp. *kefiranofaciens* subsp. nov. and *L. kefiranofaciens* subsp. *kefirgranum* subsp. nov. [10].

Some years later, the first complete genome sequence of *L. kefiranofaciens* was reported by Wang et al. [13], corresponding to strain *L. kefiranofaciens* subsp. *kefiranofaciens* ZW3, which contains a circular chromosome of 2,113,023 bp and two plasmids (pWW1 (194,769 bp) and pWW2 (46,296 bp)) (Figure 1).

## 3. Isolation and Growth Conditions

Rivière et al. [11], who were the first to isolate a capsule-forming *Lactobacillus* species from Russian grains, used MRS agar containing lactose with holes filled with kefir extract in order to imitate the growth conditions of the grain. Toba et al. [12] further reported that these capsular bacteria are difficult to isolate or need complex media. They isolated from kefir a capsule-forming homofermentative *Lactobacillus* species using a new simple medium containing milk whey, which was called kefir grain polysaccharide-producing lactobacillus (KPL) agar and they used the same medium containing deproteinized milk whey for the bacterial growth. Two years later, the same team examined four further ropy colonies isolated from kefir grains and described *L. kefiranofaciens* sp. nov. [3], which was homofermentative contrary to the reported kefiran-producing *L. brevis* described by Rivière et al. [11]. The growth of the strains was performed using a modified KPL medium [15] developed for the isolation and growth of bacteria that produce capsular polysaccharides at 30 °C in anaerobic steel wool jars at 100% CO_2_. Colonies after 10 days at 30 °C on modified KPL agar (pH 5.5) were described as circular or irregular, of 0.5 to 3.0 mm in diameter, convex, transparent to translucent, white, smooth to rough, and ropy [3]. It is worth mentioning that KPL medium is a complex chemically defined medium containing lactic acid whey, white table wine, glucose, galactose, Tween 80 and agar, therefore providing a full range of growth factors that may be required by *L. kefiranofaciens* strains. According to the authors, decreasing the amount of wine (<7% *v*/*v*), replacing 7% *v*/*v* wine with 1% *v*/*v* ethanol, or omitting Tween 80, resulted in diminished growth of the isolates. Moreover, replacement of lactic acid whey with deproteinized whey in KPL agar did not favor growth at all [12].

Additionally, Fujisawa et al. [3] used Briggs liver (BL) broth [16] for further growth at 37 °C of the isolated strains. However, it is notable that all isolates grew on modified KPL agar at 30 °C, but not on BL agar [17], which permits the growth of anaerobic bacteria, especially lactobacilli and bifidobacteria, or on MRS, which is the most common growth medium for lactobacilli. KPL (containing 140 instead of 70 mL/L of white wine) is the medium suggested by BCCM/LMG (Belgian Coordinated Collections of Microorganisms/LMG Bacteria Collection; medium number 264) for the growth of *L. kefiranofaciens* LMG strains 19149 and 19818, which correspond to the strains initially isolated by Fujisawa et al. [3]. Furthermore, lactose-digested whey (LDW) medium [12] has also been used by Fujisawa et al. [3] in an assay for acid production from carbohydrates and LAW medium (acronym not defined) by Mainville et al. [18] for the growth of lactobacilli including *L. kefiranofaciens* strains.

A few years later, Takizawa et al. used Rogosa cheese whey (R-CW) medium (pH 5.4) [19] for the isolation and growth of *Lactobacillus kefirgranum* sp. nov. and *Lactobacillus parakefir* sp. nov. strains at 30 °C and 100% CO_2_ [4]. Regarding *L. parakefir*, after five days of incubation at 30 °C, colonies were 0.5 to 2.0 mm in diameter, circular to irregular, flat, opaque, white, and rough, while *L. kefirgranum* colonies were 0.5 to 3.0 mm in diameter, circular to irregular, convex, opaque, white, and smooth to rough [4]. The use of R-CW medium has also been reported by Lemieux et al. [20] for the growth of two *L. kefiranofaciens* subsp. *kefiranofaciens* and two *L. kefiranofaciens* subsp. *kefirgranum* strains.

Furthermore, during a biodiversity study of several kefir grains and kefir fermented milk samples, the bacterial strains, which were further identified as *L. kefiranofaciens* subsp. *kefiranofaciens* or *kefirgranum*, were isolated using a milk-based medium, namely MLR, after anaerobic incubation at 30 °C [10]. Preparation of MLR medium was performed by mixing an agar solution with UHT (ultra-high temperature-treated) milk, which contained yeast extract and glucose and was acidified to pH 5.4 with acetic acid. BCCM/LMG also suggests a medium containing yeast extract, glucose and low-fat UHT milk (medium number 274) for the growth of *L. kefiranofaciens* LMG 19818.

Finally, supplemented whey and MRS media, both at pH 6.2, have been used for the isolation of ropy strains and exopolysaccharide (EPS) production, respectively [21], while MRS dissolved in lactic whey (LW-MRS) has also been reported for successful *L. kefiranofaciens* isolation [22]. Although plain MRS has not always been reliable for *L. kefiranofaciens* isolation [23,24,25], there are several authors who have reported its successful use; however, anaerobic conditions and seven days of incubation at 30 °C are necessary for bacterial growth [26,27,28].

Taking into consideration the abovementioned approaches concerning *L. kefiranofaciens* isolation and growth, it is obvious that conventional culturing techniques can be unsuccessful [29,30]. This phenomenon can be attributed to several factors, including the strictly anaerobic character and particular growth nutrient requirements of this microorganism [21], its high affinity for the grain matrix components and the symbiotic nature of the kefir microbiota [11,27,31]. Based on these characteristics and its ability to synthesize the polymeric grain’s matrix, *L. kefiranofaciens* dominates the interior of the kefir grain, while its dominance declines outside the grain. Indeed, a decline of *L. kefiranofaciens* abundance was observed within a few transfers when the microbial community was grown without the grain [32]. Moreover, it has been assumed that *L. kefiranofaciens* develops poorly without partners due to its active synthesis of the EPS kefiran and its tendency to grow in close associations [30]. Interestingly, during a study of cross-feeding interactions among the complex microbial community of kefir, it was shown that *L. kefiranofaciens* dominates the community, although it has no fitness on its own in milk and can survive in milk by cooperating with its fellow community members, such as *Leuconostoc mesenteroides* [32]. This study revealed that both species benefit from each other’s presence.

## 4. Biochemical and Physiological Characteristics

When Toba et al. [12] reported the isolation of capsule-forming homofermentative lactobacilli from kefir grains, they detected Gram-positive rod-shaped bacteria surrounded by large capsules when stained with India ink and fibrillar material adhering to the rods when examined with scanning electron microscopy (SEM). Fujisawa et al. [3] reported that *L. kefiranofaciens* strains are catalase-negative rods surrounded by capsules, as shown by India ink preparations, and do not produce gas from glucose. They reported that fermentation of sugars seemed to depend on the strain, while milk was curdled. The production of DL-lactic acid was also reported, with a marked excess of D-lactic acid, while there was no growth at 15 or 45 °C.

Emended description of *L. kefiranofaciens* by Fujisawa et al. [3] clarifies that they are Gram-positive, non-motile, non-spore-forming rods that are generally 0.5–1.2 × 3.0–20.0 μm in size and occur as single cells, in pairs or occasionally in short chains [10], while colony morphology is subspecies-dependent (see below). They are facultatively anaerobic and produce DL-lactic acid homofermentatively, while they do not produce catalase. Moreover, they do not produce gas from glucose or gluconate, nor is arginine deaminated. Milk is, however, curdled.

The fermentation profile was elucidated after a detailed description of the *L. kefiranofaciens* subsp. *kefiranofaciens* and subsp. *kefirgranum* reported by Vancanneyt et al. [10]. *L. kefiranofaciens* subsp. *kefiranofaciens* produces acid from sucrose, but not from amygdalin, arbutin, cellobiose, b-gentiobiose, inulin, salicin, trehalose or D-turanose, while acid production from N-acetylglucosamine, maltose, melibiose and D-raffinose depends on the strain. Hydrolysis of aesculin is negative. On the other hand, *L. kefiranofaciens* subsp. *kefirgranum* also produces acid from maltose and melibiose, and, for nearly all strains, also from D-raffinose, salicin, sucrose and trehalose, while acid production from amygdalin, arbutin, cellobiose, b-gentiobiose, N-acetylglucosamine, inulin and D-turanose is strain-dependent. Hydrolysis of aesculin by this subspecies is positive.

Three strains of *L. kefiranofaciens* subsp. *kefirgranum* isolated from Russian kefir grains exhibited fermentation of galactose and even trehalose but not arabinose and they hydrolyzed esculin. Neither of these strains grew at 15 °C, they did not produce gas from glucose or gluconate, nor did they produce ammonia from arginine, and produced both isomers of lactic acid [18].

Colonies of *L. kefiranofaciens* subsp. *kefiranofaciens* strains after 7–14 days of incubation at 25 or 30 °C on MLR agar were transparent, glossy, convex and extremely slimy, like those of *L. kefiranofaciens* LMG 19149^T^, while after 10 days of incubation at 30 °C on KPL agar they were circular or irregular, 0.5–3.0 mm in diameter, convex, transparent to translucent, white, smooth to rough and ropy (Figure 2A,B). On the other hand, colonies of *L. kefiranofaciens* subsp. *kefirgranum* strains were white, dry, compact, dull and bulging on MLR agar, like those of *L. kefirgranum* LMG 15132^T^, while on R-CW agar after 5 days at 30 °C they appeared to be 0.5–3.0 mm in diameter, circular to irregular, convex, opaque, white and smooth to rough. It is also worth mentioning that this subspecies forms a flocculus or powdery sediment in MLR broth and grows weakly at 15 °C [10].

Phylogeny and shotgun metagenomics sequencing have been combined with metabolomics, gas chromatography-mass spectrometry (GC-MS) and sensory analysis to link microbial species with volatile compound production in kefir beverages [34]. Strong correlations between *L. kefiranofaciens* and carboxylic acids and ketones associated with cheesy flavors, as well as esters associated with fruity flavor, were revealed. Similar results were obtained by Dertli and Çon [35]. Additionally, Walsh et al. [34] showed that adding *L. kefiranofaciens* NCFB 2797 to kefir resulted in increasing the levels of 2-heptanone and esters.

## 5. Species and Subspecies Discrimination

### 5.1. Culture-Dependent Approaches

Useful tools for discrimination of lactobacilli, including *L. kefiranofaciens* isolated from kefir and other dairy products using culture-dependent approaches, comprise either phenotype based methods, e.g., whole cell protein electrophoretic profiles (by SDS-PAGE) and whole bacteria compounds (by Fourier-transform infrared spectroscopy; FT-IR), or molecular techniques, such as random amplified polymorphic DNA (RAPD), sequence-based identification using phenylalanyl-tRNA synthase gene (*pheS*), repetitive element palindromic PCR fingerprinting (rep-PCR) with the (GTG)_5_ primer, and 16S rRNA gene sequencing (reviewed by Bengoa et al., 2018) [36]. Recently, colony PCR with *L. kefiranofaciens* species-specific primers (Table 1) has also been performed [28].

### 5.2. Discrimination at the Subspecies Level

Discrimination of *L. kefiranofaciens* at the subspecies level is not, however, possible using genotypic approaches since both *L. kefiranofaciens* subspecies share 100% 16S rRNA gene sequence similarity [10]. The molecular typing at the subspecies level also appeared to be impossible using *pheS* gene sequencing or (GTG)_5_-fingerprinting [27]. Additionally, although strain variations within the *L. kefiranofaciens* species were shown with the restriction fragment length polymorphism (RFLP) method using *Hin*dIII as the restriction endonuclease, subspecies discrimination was not achieved [18]. The authors reported that a polyphasic characterization of the LAB in kefir by combining genotypical, phenotypical and biochemical methods, proved to be a valuable tool for typing at the strain level. These results were in accordance with those of Takizawa et al. [39], who reported no genotypic differences between the two subspecies. Mainville et al. [18] also confirmed the possibility that the main difference between the two subspecies *kefiranofaciens* and *kefirgranum* is the EPS production that is only carried out by the subspecies *kefiranofaciens* [18]. This could be attributed to the loss of a plasmid coding for the slime-producing trait by the *kefirgranum* subspecies [40,41]. 

Nevertheless, differentiation of *L. kefiranofaciens* at the subspecies level can be achieved by whole protein profile [10,27,39] and FT-IR analysis [42]. Interestingly, the whole-cell protein profiles of the subsp. *kefirgranum* strains can be differentiated visually by the varying position of a dominant protein band with a molecular mass of 38–60 kDa [10]. The authors reported that these strain-specific variable dense bands could indicate the presence of a surface (S)-layer, as it was previously demonstrated for other species of the *Lactobacillus acidophilus* group [43].

### 5.3. Culture-Independent Approaches

On the other hand, the total microbial structure of products having a complex microbiota, such as kefir, may be inaccurately described by culture-dependent techniques and conventional molecular methods, as only dominant populations are identified, which may not necessarily play important roles in the overall community dynamics [31]. Therefore, culture-independent approaches have been used, with the PCR-denaturing gradient gel electrophoresis (PCR-DGGE) being the most widely applied in kefir [26,27,28,29,44,45,46,47,48]. However, despite differences observed between PCR-DGGE patterns, the band migration in both subspecies is similar, and thus subspecies discrimination cannot be achieved [27]. A species-specific PCR amplification method has been developed to allow the detection of *L. kefiranofaciens* in different tissues using primers designed from unique *L. kefiranofaciens* DNA sequences (Patent US 2009/0130.072 A1, Table 1) [20]. These primers are reported to be highly specific, being able to detect *L. kefiranofaciens* DNA in samples isolated from feces, colon content, mucosa and whole colon.

In addition, various LABs were identified within the consortium of a Belgian kefir grain by 16S rRNA gene variable region sequencing, among them *L. kefiranofaciens* at the subspecies level [49]. In this case, specific primers either for the dominant or the less abundant bacterial groups were used separately for DNA amplification and the amplified fragments were sequenced after being cloned in *Escherichia coli*. Similar experiments have been performed with Brazilian kefir grains [22]. Furthermore, Kim et al. [37] designed a novel real-time PCR primer and probe set for the rapid detection of *L. kefiranofaciens* and studied kefir grains and beverages using a real-time PCR assay for the first time (Table 1). Wang et al. [28] reported the use of fluorescence in situ hybridization (FISH) and real-time quantitative PCR (RT-qPCR), demonstrating that *L. kefiranofaciens* was the only dominant bacterial species in Tibetan kefir grains (Table 1). Recently, multiplexed qPCR assays, using dual-labeled oligoprobes of TaqMan assay, which anneal specifically to a target region, were developed in order to specifically detect and quantify several microorganisms in kefir grains and beverages’ microbial communities and consequently evaluate their population dynamics and microbial interactions [38]. Primer-probe sets targeting species-specific genes were designed for six bacteria and five yeasts, among them *L. kefiranofaciens* (Table 1). The target gene for *L. kefiranofaciens* was the gene encoding the DNA helicase RecG and the probe was labeled with the fluorescent dye Q705 at 5′ end, and the corresponding quencher BHQ3 at 3′ end.

### 5.4. High-Throughput Sequencing, Metabolomics and Transcriptomics

Finally, an automated high-throughput sequencing technique, such as 16S or 26S rRNA gene pyrosequencing, which can allow the identification of bacteria and yeasts that are present even in small abundances and are rarely associated with the microbial community of complex ecosystems, has been successfully applied to identify *L. kefiranofaciens* [23,28,46,50]. The best has already come with whole genome shotgun pyrosequencing, as this approach does not involve cloning or 16S rRNA gene amplification and overcomes the aforementioned problems involved with alternative identification methods, discriminating *L. kefiranofaciens* at the sub-species level [51]. In 2016, Walsh et al. [34] combined for the first time whole-metagenome shotgun sequencing with metabolomics to link microbial species with volatile compound production in kefir. The authors also revealed that in the early stages of fermentation *L. kefiranofaciens* was the dominant species, whereas *L. mesenteroides* prevailed at the latter stages. However, in complex food microbial ecosystems, such as kefir, when bacteria are identified by metagenomic analyses at the species level, among them *L. kefiranofaciens*, there is always a hazard lurking: the possibility that population dynamics are skewed if there are dead cells present [52]. Large numbers of dead cells may indicate the importance of a species for the microbial community; nevertheless, culture-dependent methods are necessary for pinpointing which species are actively involved. Moreover, during recent years, the development of various technologies, including hybridization- or sequence-based approaches, have permitted the deduction and quantification of transcriptomes [53]. RNA sequencing (RNA-Seq) is a revolutionary approach to transcriptomics, providing both mapping and quantification [53], and has recently been used to study the expression of genes linked to the growth and metabolism of *L. kefiranofaciens* [32].

## 6. Ecological Niches

As already mentioned, *L. kefiranofaciens* was first isolated from kefir grains [3]. *L. kefiranofaciens*, along with the highly biofilm-forming *L. kefiri*, is one of the key LAB species in kefir grain ecosystems, being involved in the mechanism of grain formation due to its strong auto-aggregation capacity, very high hydrophobicity, and positive cell surface charge at pH 4.2 [54]. Additionally, Wang et al. [54] reported that lactobacilli arranged in short chains, such as *L. kefiri*, occupy the kefir grain surface, while lactobacilli arranged in long chains, such as *L. kefiranofaciens*, aggregate towards the center, as revealed by SEM.

Since the first report of Fujisawa et al. [3], a plethora of strains have been isolated from kefir grains using various culture-dependent methods (Table 2). Additionally, various culture-independent methods have been applied in order to verify the occurrence of the species in kefir (Table 2), since conventional culturing approaches do not often succeed in identifying the species [27]. Thereafter, all over the world, *L. kefiranofaciens* has been detected and often isolated from kefir grains in Argentina, Belgium, Brazil, China, France, Germany, Greece, Ireland, Italy, Lithuania, Malaysia, Korea, Russia, Slovenia, Taiwan, Turkey, the United Kingdom and the USA (Table 2).

Recently, it has been shown that the microbiota of kefir grains produced in China, Germany and the US was stable after sub-culturing in goat milk for 2 to 4 months; *L. kefiranofaciens* was one of the species detected in the samples using metagenomics analysis [65]. Interestingly, *L. kefiranofaciens* was revealed to be the only dominant and stable bacterial species in Tibetan kefir grains that had been cultured continuously for 10 months, either naturally or aseptically, regardless of culture conditions and time of cultivation [28]. Moreover, Wang et al. [28] showed that the species exhibited two distinct morphotypes of short and long rods (3.0 μm and 10.0 μm in length, respectively) when colonizing either the outer surface or interior of the grains. Therefore, the authors provided evidence for the trophic adaptation of the cells to the hollow globular grain structure. It is also possible that physiological stages or external stresses, such as cultivation conditions and limitation of available nutrients, can influence cell sizes and chain lengths [67].

It is notable that *L. kefiranofaciens* was not detected in several kefir beverages, which were examined in parallel with their respective grains [29,30], likely due to its low abundance, the high detection limit of the method used or the high affinity of the bacterium to the grain matrix [37,58]. It is also important that colonies corresponding to *L. kefiranofaciens* may be present only in the low dilution plates, which show confluent colonies, and not in the high dilution plates, which show countable colonies [23]. One more possible reason explaining the low or zero detectability of *L. kefiranofaciens* in kefir beverages is its outgrowth during the fermentation by other bacterial species, even if it was dominant in the beverage at the early stages. Outgrowth of *L. kefiranofaciens* after 8 h of fermentation by *L. mesenteroides* has already been reported by Walsh et al. [34]. There are, however, multiple references reporting the detection of *L. kefiranofaciens* and sometimes its isolation from kefir beverages as well [10,34,37,47,58].

Lately, the microbiota of kefir grains and beverages produced in Greece were examined using a holistic approach combining classical microbiological, molecular and amplicon-based metagenomics analyses [68]. Although *L. kefiranofaciens* was not isolated using the growth media MRS and Rogosa, the amplification of the V1–V3 hypervariable region of the 16S rRNA gene revealed its occurrence in both kefir grains and beverages (unpublished data). Due to the high-level similarity between closely related taxa, these results were not reported, and the microbiota of the samples was evaluated up to the genus level for a more accurate identification, although the discriminatory power for bacterial identification is offered by the V1–V3 region of the 16S rRNA gene, even at the species level [69]. In 2020, Kalamaki et al. also reported the detection of *L. kefiranofaciens* in two Greek kefir grain samples using 16S rRNA gene sequencing [62].

Moreover, strains of *L. kefiranofaciens* have been isolated from the Mongolian hurunge, manufactured by nomadic families in Inner Mongolia using roasted millet and fresh cow, mare, or camel milk [70]. Strains of *L. kefiranofaciens* have also been isolated from Mongolian koumiss, the traditional fermented mare milk beverage [71,72]. Watanabe et al. [73] reported the isolation of *L. kefiranofaciens* strains from two traditionally fermented dairy products of Mongolia, namely airag and tarag. According to their results, the microbial diversity of these products was affected more by the milk type rather than the geographical origin.

*L. kefiranofaciens* occurrence and identification has also been reported in 10 samples of koumiss collected in Xinjiang Uygur Autonomous Region in China by PCR-DGGE [74]. There is one more study reporting the detection of *L. kefiranofaciens* in koumiss and raw mare milk collected in Xinjiang, China, using PacBio single-molecule real-time (SMRT) sequencing to profile full-length 16S rRNA genes [75]. Interestingly, the authors reported that the raw milk bacterial microbiota diversity was more complex and diverse than that of koumiss. *L. kefiranofaciens* was also presumably detected by pyrosequencing in home-made yogurt samples in Xinjiang, China [76], as well as in airag, khoormog and tarag [77]. It is worth mentioning that khoormog is made mainly from camel milk and *L. kefiranofaciens* was one of the dominant species in this product. On the other hand, *L. kefiranofaciens* was barely present in 17 tarag samples collected from various regions of Mongolia and China and analyzed with pyrosequencing, leading to the conclusion that geographical origin may influence the microbial biodiversity [78]. Additionally, *L. kefiranofaciens* was detected in Rushan cheese samples produced in Yunnan province in China using 16S rRNA gene sequencing [79] and in a Camembert-type cheese made in Shanghai, China, by PCR-DGGE, and strains were isolated using acidified MRS containing 0.4 mg/mL nystatin after growth at 37 °C for 48 h anaerobically [80].

Finally, during a study comparing the microbiota of the traditional doogh and industrial yogurt samples of Iran, *L. kefiranofaciens* was detected using PCR-DGGE and RT-PCR [81]. Doogh is a fermented milk drink obtained by diluting yogurt with drinking water, addition of salt and heat treatment, therefore commercial starters are used for its industrial production. On the contrary, traditional doogh is a buttermilk occurring as a byproduct of butter production from yogurt in goatskin or musk bags. Sayevand et al. reported collecting the home-made samples directly from the skin bags or the containers used for storing [81].

## 7. EPS Production–Kefiran

The cell walls of Gram-positive bacteria usually contain polysaccharides along with ‘accessory polymers’, such as teichoic acids, teichuronic acids and proteins [82]. Several LAB excrete EPS of elevated molecular weight and particular physical and rheological properties, therefore they are suitable as viscosifying, stabilizing, gelling or emulsifying agents. As LAB EPS are produced by GRAS bacteria, they are promising for the generation of new food thickeners [21].

### 7.1. Kefiran Production and Purification

One of the major polysaccharides participating in the kefir grain assembly is the water-soluble glucogalactan called kefiran, described by Rivière et al. in 1967 [11]. It is produced by *L. kefiranofaciens* subsp. *kefiranofaciens* strains and is known for the formation of viscous colonies (Figure 2A) [83,84,85], whereas the production level is strain-dependent [27] and affected by the fermentation medium/conditions [86]. Indeed, kefiran production has been optimized (58.02% increase) using a growth medium containing sucrose, yeast extract and KH_2_PO_4_ in a semi industrial-scale bioreactor [87]. In an early study on the cell capsular polysaccharide accessory polymer, the low yield of teichoic acid suggested that kefiran is the main accessory polymer in the cell-wall of *L. kefiranofaciens* [82]. The EPS production capability of *L. kefiranofaciens* is probably responsible for its participation in the formation of kefir grains matrix, as well as the viscosity of the final kefir beverage [30]. Additionally, kefiran-producing and capsulated *L. kefiranofaciens* are located all over the kefir grain and increased towards the center, while some non-kefiran producing *Lactobacilllus* species populated only a small region at the surface [88]. Kefiran can be purified using the method of Piermaria et al. [89] by precipitation with pure cold ethanol after growth of *L. kefiranofaciens* in a growth medium containing lactose, yeast extract, KH_2_PO_4_, sodium acetate, triammonium citrate, MgSO_4_ and MnSO_4_. A detailed procedure for the isolation of kefiran is also described by Zajsek et al. [90] using a trichloroacetic acid solution for exclusion of proteins and chilled acetone for kefiran precipitation.

It is important to mention that *L. kefiranofaciens* growth and capsular kefiran production are enhanced in a mixed culture with *Saccharomyces cerevisiae* [84,91,92]. *S. cerevisiae* does not utilize lactose but under aerobic conditions it assimilates lactic acid, which inhibits LAB growth when accumulated, and results in a decrease in useful metabolites associated with growth, such as kefiran. [84]. In addition, kefiran production is enhanced in a mixed culture mainly because of the physical contact of *L. kefiranofaciens* with *S. cerevisiae* [91]. Furthermore, it has been reported that kefir grain formation begins with the self-aggregation of *L. kefiranofaciens* and *Saccharomyces turicensis* by forming small granules and co-aggregation increases when *S. turicensis* and kefir LAB strains (*L. kefiranofaciens* and *L. kefiri*) are co-cultured [54]. The importance of CO_2_ release in kefir by heterofermentative lactobacilli is also highlighted, since CO_2_ contributes to the creation of an anaerobic environment, as well as to the overall taste of kefir [48]. Consequently, the improvement of the net quantity of kefiran highlights the importance of the symbiosis in the kefir consortium. Finally, it is notable that a growth medium for *L. kefiranofaciens* has been developed using rice hydrolysate (RH) for large-scale kefiran production [93].

Although at present kefiran production is attributed to *L. kefiranofaciens* subsp. *kefiranofaciens*, it has been suggested that lactobacilli other than *L. kefiranofaciens* are gifted with this asset [94]. More specifically, Frengova et al. reported EPS production using as starters an association of *Lactobacillus delbrueckii* subsp. *bulgaricus* (*L. bulgaricus*) with *Streptococcus thermophilus*, *Lactococcus lactis* subsp. *lactis*, *L. helveticus* and *S. cerevisiae*, and identified the kefiran as an EPS containing glucose and galactose at a 1.0:0.94 ratio [94].

Back in 1990, gel filtration chromatography was used by Yokoi et al. to determine the molecular mass of kefiran, and two peaks were found corresponding to molecular masses of 1.0 × 10^6^ and 4.0 × 10^6^ [95]. After the work of Yokoi et al., various values have been reported in the literature ranging between 5.5 × 10^4^ and 1.0 × 10^7^ Da, depending on the extraction conditions and the degradation that may occur during this stage, while recently Pop et al. reported that the molecular mass of kefiran ranges between 2.4 × 10^6^ and 1.5 × 10^7^ Da [96]. Molecular mass also depends on the composition of the fermentation medium, as highlighted by Wang and Bi [86].

### 7.2. Kefiran Chemical Structure

Kefiran contains approximately equal amounts of glucose and galactose [97] and comprises at least 24% to 25% (*m*/*m*) of the kefir grains dry matter. Interestingly, variations in the composition of the *L. kefiranofaciens* fermentation medium provoke changes in the kefiran chemical structure [86,96]. Wang and Bi showed that using maltose as the sole carbon source resulted in a 1:10 glucose/galactose molar ratio of kefiran and a maximum viscosity of 73.86 ± 5.3 dL/g [86]. Pop et al. also analyzed the composition of monosaccharides in kefiran by high-performance liquid chromatography (HPLC) analysis and reported that it is composed of glucose and galactose at a relative molar ratio of 0.94:1.1 [96]. Additionally, FT-IR spectroscopy revealed that kefiran is composed of an α- and β-configuration of monosaccharides in the pyranose form [96]. The peaks detected and the ring vibrations of the FT-IR spectra of purified kefiran indicated the presence of glucose, galactose, and β-linkages, thus verifying the results of Piermaria et al. [89].

Kefiran was characterized by means of viscosity, optical rotatory power, circular dichroism, and IR spectroscopy [98]. Kooiman was the first to elucidate the chemical structure of kefiran extracted from kefir [97], and Mukai et al. further examined the structure of kefiran produced by *L. kefiranofaciens* strain K1 [83,99]. Kefiran isolated from kefir grains has a backbone composed of glucose and galactose [100]. The structure corresponds to a branched hexa- or heptasaccharide repeating unit that is itself composed of a regular pentasaccharide unit, to which one or two sugar residues are randomly linked (Figure 3) [101]. Linkages of kefiran cannot be hydrolyzed by the digestive enzymes of the human gastrointestinal tract; on the contrary, kefiran can be degraded by members of the gut microbiota [102].

### 7.3. Genomics Studies

In 2011, the complete genome sequence of *L. kefiranofaciens* ZW3 revealed that one of the most significant features of the strain is its ability to produce high-yield EPS [13]. A 14.4-kb EPS gene cluster is present containing 17 EPS-related genes, which show high similarity to the genes of enzymes involved in EPS regulation, polymerization, chain length determination and export. Moreover, 12 of these genes have homologies with other *Lactobacillus* species, while the remaining five genes seem to be unique in the ZW3 genome and are probably responsible for key enzymes to produce unique EPS. Recently, functional and bioinformatics analysis of an EPS-related gene (*epsN*) from *L. kefiranofaciens* ZW3 was performed [103]. It was shown for the first time that EpsN has a functional property critically affecting *L. kefiranofaciens* EPS biosynthesis.

A comparative genomics study, including *L. kefiranofaciens* ZW3, showed the presence of a series of genes relevant to dairy environment and the human and animal gastrointestinal tract, among them genes responsible for EPS production [104]. Multiple copies of enzymes related to lactose and galactose catabolism to permit full nutrient use in a dairy environment were initially found. The metabolic pathways in ZW3 were further investigated using the KEGG database by exploring the Leloir pathway, which is related to EPS production in LAB. It was found that the monosaccharide composition of EPS consists of mannose, galactose and glucose and enzymes associated with UDP-glucose, UDP-galactose and GDP-mannose, including glucose-6-phosphate isomerase (pgi), α-phosphoglucose mutase (pgm), UDP-glucose pyrophosphorylase (ugp), UDP-galactose 4-epimerase (uge) and mannose-6-phosphate isomerase (mpi). These enzymes were differently expressed in the two different growth media used (modified MRS and whey medium) during growth. The carbon flux is regulated through the EPS synthetic pathway by four enzymes encoded by the *pgm*, *ugp*, *uge*, and *pgi* genes, and, in turn, it affects EPS yield (Figure 4). As expected, the activity of the enzymes involved in the EPS synthesis pathway were affected by different components of the growth medium [104].

### 7.4. Applications

Kefiran can be used as a stabilizer, emulsifier, thickener, gelling agent and fat substitute [84]. The rheological properties of chemically acidified skim milk gels are enhanced by kefiran and their apparent viscosity, as well as their storage and loss modulus, are increased. Therefore, it can be used as a food-grade additive for fermented products [21,105]. The physicochemical properties, such as the thermal stability, emulsifying capability and flocculating activity, of the heteropolymeric EPS of glucogalactan nature produced by *L. kefiranofaciens* strain ZW3 studied by FT-IR spectroscopy and GC analysis revealed that it exhibits higher emulsifying capability compared to commercially available hydrocolloids like xanthan gum, guar gum and locust gum [21]. A detailed overview of its multifarious applications in the agri-food and biomedical sectors has recently been published [106].

## 8. Safety Status of *L. kefiranofaciens*

Species of the *Lactobacillaceae* family are probably those most widely used as starter or adjunct cultures for food applications (e.g., fermented products) due to their long history of safe and technological use [107]. In this context, a strong argument for the safety of lactobacilli isolated from kefir, including *L. kefiranofaciens* strains, is the fact that no pathogenicity and/or toxicity has been associated with kefir consumption over the years [20]. However, studies for evaluating the safety status of specific *L. kefiranofaciens* strains have been conducted.

The acute oral toxicity of *L. kefiranofaciens* M1, a strain that can adapt to heat, cold, acid and bile salt stress [108], was evaluated in rats [109]. Animal body weight measurements, hematology and blood serum biochemical tests, organ histopathology and urine analysis were performed for animals receiving three different strain doses (i.e., low, medium and high dose) for 28 days. The results obtained indicated that no adverse and/or toxicity effects were detected for all strain doses tested. Consequently, the highest dose used in the study (1.8 × 10^10^ cfu of *L. kefiranofaciens* M1 per kg of bodyweight) was considered the no-observed-adverse-effect-level (NOAEL) for the tested animals. Moreover, no cytotoxicity effect on 3T3-L1 adipocytes was observed in a study evaluating the effect of the above strain on adipocyte differentiation and key lipogenesis markers [110].

To demonstrate the safety of *L. kefiranofaciens* DN1, Jeong et al. [111] assessed the strain’s hemolytic activity on blood agar and its gelatinase activity. DN1 strain exhibited no growth or hemolytic activity on blood agar and no gelatinase activity, contributing to the conclusion that *L. kefiranofaciens* DN1 should be considered safe in vivo.

Some years ago, a malleable protein matrix (MPM), containing whey proteins, peptides, a proprietary *L. kefiranofaciens* strain (ES R2C2), exopolysaccharides and calcium was produced by an innovative industrial process for whole whey fermentation [112,113], and during the subsequent years many studies have been performed using this specific product. A human study evaluated the lipid-lowering properties of MPM in patients with hypercholesterolemia. Both safety and tolerability were assessed by recording adverse events (AE) as well as by measuring vital signs and biochemical and hematological variables, and it was reported that the MPM product was tolerated well without severe AE in subjects [114].

Interestingly, although kefir is consumed by millions of people around the globe with no AE reported so far, the literature contains sparse information on the safe levels of kefir intake in animal trials [115]. A 4-week kefir administration in rats, either with a normal (0.7 mL/day/animal) or a high (3.5 mL/day/animal) dose, did not show harmful effects on animals as determined by growth, hematology, blood chemistry or potential pathogenicity analyses in tissues [116]; however, the microbial composition of kefir was not determined to correlate its safe use with the presence of *L. kefiranofaciens*. Additionally, administration of kefir, with *L. kefiranofaciens* as the most abundant species, improved the survival rate in a fly model for Alzheimer’s disease without any side effects [117].

## 9. Functional and Probiotic Properties of *L. kefiranofaciens*

As recently reviewed by Slattery et al. [118], a considerable number of studies have been performed focusing on the probiotic properties and health benefits of bacterial species dominating kefir products. Indeed, as *L. kefiranofaciens* represents a significant proportion of the *Lactobacillus* species found in kefir, it has been extensively studied for its impact on human health.

### 9.1. Antimicrobial Activity

Regarding the antimicrobial properties of *L. kefiranofaciens*, the prophylactic and therapeutic potential against enteric bacterial pathogens have been studied both in vitro and in vivo. Specifically, in vitro experiments indicated that *L. kefiranofaciens* isolated from kefir grains managed to inhibit enteropathogenic bacteria used as indicators [22,119]. Moreover, *L. kefiranofaciens* CYC 10058 exhibited antimicrobial activity against a few enteropathogenic bacteria and inhibited *Salmonella typhimurium* attachment to Caco-2 cells [120]. The effects of *L. kefiranofaciens* M1 on enterohemorrhagic *E. coli* (EHEC) infection using intestinal cell models and mice was also investigated. Strain M1 had a protective effect on Caco-2 intestinal epithelial cells as limited EHEC-induced cell death and a reduced loss of epithelial integrity were observed. In vivo, strain M1 administration in mice resulted in prevention of the infection-induced symptoms, intestinal and renal damage, bacterial translocation and Shiga toxin penetration with possible mechanisms proposed for the enhancement of mucosal immunity and intestinal barrier functionality [121]. Another *L. kefiranofaciens* strain, namely DN1, alone or combined with the yeast *Kluyveromyces marxianus*, prevented *Salmonella* Enteritidis colonization when administrated to chickens, with the most promising results being observed after early administration (before chicken infection with *Salmonella*) of the bacterium–yeast combination [122]. A bactericidal effect was also exhibited by an EPS produced by *L. kefiranofaciens* DN1 against *Listeria monocytogenes* and *Salmonella* Enteritidis [123]. However, the results of high-performance size-exclusion chromatography (HPSEC) indicated that the EPS produced by DN1 was not kefiran, but a novel bioactive compound. Furthermore, except antimicrobial activity against various food-borne pathogens, *L. kefiranofaciens* strains isolated from Turkish kefir grains also exhibited antifungal activity against food-spoilage species, such as *Alternaria alternata*, *Aspergillus paraciticus* and *Fusarium oxysporum* [61]. Finally, the ability of *L. kefiranofaciens* kefir isolates to survive in an experimental oral environment was tested, along with their antimicrobial and anti-biofilm activities against the main causal pathogens of early dental caries *Streptococcus mutans* and *Streptococcus sobrinus* with promising results as potential oral probiotics [124].

### 9.2. Immunomodulatory Action

Several investigations and in-depth studies have established, with both clear and unclear mechanisms, immunomodulatory strain-dependent probiotic actions. *L. kefiranofaciens* M1 showed in vitro immunomodulatory properties by regulating the production of a number of pro-inflammatory cytokines, probably through the Toll-Like Receptor 2 (TLR-2) pathway, in a murine macrophage cell line model [125]. The immunomodulatory, and more specifically the anti-allergic properties of the strain, were further established when heat-inactivated M1 cells effectively inhibited immunoglobulin (Ig)-E production in ovalbumin-sensitized Th2-polarized mice due to a skewed Th1/Th2 immune response toward Th1 dominance and elevated CD4^+^CD25^+^ regulatory T cells [126]. Moreover, all features of the asthmatic phenotype in mice, including specific IgE production, airway inflammation, and development of airway hyperresponsiveness, were depressed in a dose- and time-dependent manner after treatment with heat-inactivated M1 cells [127].

Except anti-allergic potential, *L. kefiranofaciens* strains have also been investigated as an alternative therapy for intestinal disorders both in vitro and in vivo. In vitro results indicated that *L. kefiranofaciens* M1 strengthened the epithelial barrier function and significantly upregulated the chemokine CCL-20 level in Caco-2 cells [128]. In the same study, M1 could ameliorate chemically induced colitis as a significant reduction in the bleeding score while colon length shortening was observed with IL-10 playing a key role in the attenuation of inflammatory responses. Further experimentation in in vivo models using germ-free mice showed that the M1 strain failed to colonize the animals and continuous consumption might be necessary to maintain its probiotic action [129]. Another *L. kefiranofaciens* strain, namely DN1, successfully altered the gut microbiota and fecal quality in mice, suggesting a constipation-alleviating effect [111]. Furthermore, a 2-week continuous oral administration of *L. kefiranofaciens* XL10 in mice modulated gut microbiota in the tested animals. Interestingly, butyric acid-producing bacteria increased, which are considered important for the intestinal barrier function and anti-inflammatory beneficial effects in the gut [130]. Finally, *L. kefiranofaciens* KCTC 5075 produces extracellular vesicles (EV), which appear to be important mediators of cell-to-cell interaction and can potentially be used for developing innovative strategies for alleviating inflammatory bowel disease. The role of EV in modulating inflammation responses has been proposed via reducing the production of inflammatory cytokines in tumor necrosis factor-α (TNF-α)-induced inflammation in Caco-2 cells in vitro and via alleviating body weight loss and rectal bleeding in a chemically-induced colitis murine model [131].

### 9.3. Role in Metabolic Disorders

The role of probiotics in metabolic disorders, such as obesity and diabetes, has been extensively reported, and, in this context, *L. kefiranofaciens* strains have also been studied for beneficial effects. Lin et al. [132] reported that *L. kefiranofaciens* M1 enhanced body weight gain when orally administered in mice receiving a high-fat diet (HFD). When the authors tried to elucidate the obesity effect of the M1 strain in comparison to the anti-obesity effect of *Lactobacillus mali* APS1, they highlighted the importance of the tripartite relationship among the host, microbiota, and metabolites for differences in inflammatory biomarker expression, and gut microbiota were reported for both lactobacilli interventions in the tested animals [110]. Moreover, the oral administration of *L. kefiranofaciens* strain M could alleviate the progression of type-1 diabetes (T1D) symptoms in streptozotocin-induced T1D murine model by stimulating the production of glucagon-like peptide-1 (GLP-1), regulating the immune-modulatory reaction and modifying the gut microbiota [133]. In another study, the effects of kefir, with *L. kefiranofaciens* species abundance, on endothelial cells and vascular responsiveness were studied in spontaneously hypertensive rats with kefir treatment being able to improve endothelial function in the tested animals [134].

### 9.4. Gut Microbiota Modulation

The distribution and colonization potential of *L. kefiranofaciens* ZW3, as well as its capacity to modulate gut microbiota in mice have been evaluated [135]. ZW3 was found to successfully adhere to and colonize the mouse gut and the profiling analysis of gut microbiota supported the function of ZW3 in beneficially altering mice intestinal microbiota. Interestingly, dietary supplementation with *L. kefiranofaciens* ZW3 improved depression-like behavior in stressed mice by modulating gut microbiota as the probiotic strain was present in the whole intestine, even seven days after its administration was stopped [136].

### 9.5. Other Health-Promoting Properties

Tibetan kefir grains dominated by *L. kefiranofaciens* species were proposed as a new alternative for biological detoxification of mycotoxins, such as ochratoxin A (OTA), in foods. In fact, an analysis of the OTA detoxification mechanism in milk revealed that both adsorption and degradation activities were exhibited by kefir grains [66]. Experiments were also performed to investigate the ability of kefir, again with the dominant *L. kefiranofaciens* species, to reduce wound area in an in vitro scratch assay with positive results [137]. Moreover, in the same study, beneficial effects of kefir incorporated into silver sulfadiazine cream were evaluated on burn wounds in vivo and it was found that kefir enhanced migration and proliferation of fibroblasts and improved fibrous connective tissue formation in the wound area of rats. Finally, kefir, with *L. kefiranofaciens* as its most abundant species, improved the climbing ability, survival rate and neurodegeneration index of flies in a *Drosophila melanogaster* model for Alzheimer’s disease (AD) [117].

### 9.6. Health-Promoting Properties of Fermentation Products or Metabolites

Apart from the use of specific *L. kefiranofaciens* strains, experiments have also been performed to evaluate the health-promoting properties of certain fermentation products or metabolites of the species. More specifically, a study using the aforementioned MPM product reported that it stimulates the innate immune defense in healthy animals, exhibits an anti-inflammatory effect in an atopic dermatitis model and reduces neutrophil infiltration associated with the inhibition of IL-1β, IL-6, and TNF-α production in a murine air pouch model [138,139,140]. The potential of MPM to regulate dyslipidemia was also investigated, focusing on blood glucose management (hamsters and fructose-fed rats) and hypertension in spontaneously hypertensive (SHR) rats [141]. It was revealed that MPM has beneficial effects on lipid metabolism, blood glucose control, and hypertension, and may contribute to the management of metabolic syndrome and cardiovascular diseases. Additionally, a similar concentrated whey deriving ingredient mixture, obtained through a fermentation process using *L. kefiranofaciens* (strain not reported), had significant triglyceride (TG) lowering properties in human subjects with combined hypercholesterolemia and higher TG levels [114]. Finally, *L. kefiranofaciens* 1P3, in the presence of sucrose, produces alpha-glucans, which exhibit interesting immunological properties as they had a significant effect on the expression of the intestinal IgA + B cells in mice, while they had low or non-existent in vitro cytotoxicity, especially in non-tumor cells [33].

### 9.7. Genomics for Probiotic-Associated Traits

Recently, high-throughput sequencing (HTS) technologies have also been implemented in studies regarding the probiotic potential of *L. kefiranofaciens*. Using whole-metagenome shotgun sequencing of three kefir samples from France, Ireland and the UK, the gene family table was inspected for genes associated with probiotic functionalities to better understand the basis of its health benefits [34]. It was shown that in three samples *L. kefiranofaciens* contained genes encoding EPS synthesis, bile salt transporter, adhesion and mucus binding proteins, as well as the type III bacteriocins helveticin, and enterolysin A, providing molecular evidence for the relevant probiotic properties of the species. Comparative genomics of *L. kefiranofaciens* ZW3 revealed adaptations to dairy and gut environment, as a series of genes relevant to these environments were identified, particularly those encoding extracellular EPS production [104]. Moreover, ZW3 tolerated pH 3.5 and 3% *w*/*v* bile salt and retained cell surface hydrophobicity and auto-aggregation, indicating its potential utilization in both the dairy industry and probiotic applications.

### 9.8. L. kefiranofaciens Patent

In 2009, a patent was filed regarding the use of a five *L. kefiranofaciens* strain consortium as a probiotic preparation in association with a suitable carrier depending on the administration manner, oral, rectal or vaginal (Lemieux et al., Patent US 2009/0130.072 A1) [20]. The consortium comprises *L. kefiranofaciens* subsp. *kefiranofaciens* strains R2C2, INIX (ATCC 43761) and ES1 and *L. kefiranofaciens* subsp. *kefirgranum* strains K2 and BioSp. The probiotic potential of the consortium refers to effects on intestinal health, immunomodulation and obesity-associated problems, such as control of blood lipid levels, hypertension and body weight, as well as protection against diabetes and tumors. Methods are provided by the patent for protecting and treating a subject against various diseases and syndromes by defining the effective amounts to be used in association with a suitable carrier to be properly formulated for oral, rectal, or vaginal administration.

## 10. Functional Properties of Kefiran

Due to its unique physicochemical and rheological properties, kefiran *per se* offers a wide spectrum of applications in the food industry by acting either as a biodegradable and edible coating and packaging material or as a texturing agent with important emulsifying and gelling effects [101,142]. In any case, it should also be noted that kefiran has been associated with promising bioactive properties, including antimicrobial, immunomodulating, anti-hypertensive and anti-tumor activities; thus, it can be considered suitable for both food and pharmaceutical applications with health-promoting benefits [106].

### 10.1. Antimicrobial Activity

Kefiran extract has shown antimicrobial activity against *Candida albicans* and several bacterial species with the highest activity against *Streptococcus pyogenes* using an agar diffusion method [143]. Moreover, when the bactericidal properties of kefiran were studied using a live/dead staining protocol assessing bacterial viability, antimicrobial activity against *Pseudomonas aeruginosa*, *E. coli*, *Staphylococcus aureus*, and *S. typhimurium* strains was observed [144]. Interestingly, when the interaction between kefiran and biomembranes was studied, the proposed mechanisms of kefiran targeting microbial cells involved disruption of the cell membrane through pore formation and detergent-like effects [145]. Finally, regarding intestinal infections, the ability of kefiran to antagonize cytopathic effects triggered by *Bacillus cereus* on Caco-2 cells was evaluated, and its ability to diminish eukaryotic cell necrosis, F-actin disorganisation and microvilli effacement was reported [146].

### 10.2. Immunomodulatory Action

Additionally, kefiran can act as an immunomodulatory molecule for the human immune system and a number of studies have focused on this research area. In fact, the in vitro anti-inflammatory action of kefiran was established for the pretreatment of mast cells with kefiran suppressing degranulation and cytokine production in a dose-dependent manner and the authors suggested that kefiran could be useful for the prevention and treatment of allergic diseases mediated by mast cells [147]. Vinderola et al. [148] reported that oral administration of kefiran induced a gut mucosal response in mice, and more precisely, it enhanced the IgA production in both the small and large intestine, and, through the cytokines released, it regulated an immune response, contributing to intestinal homeostasis. Similarly, orally administrated kefiran in mice increased the number of IgA+ cells and macrophages in small and large intestine lamina propria and peritoneal cavity, suggesting its use in intestinal pathologies [149]. Additionally, Rodrigues et al. [150] examined the anti-granuloma and anti-oedematogenic effect of kefiran extract in rats using carrageenan, dextran and histamine as stimuli and highlighted a possible anti-inflammatory effect.

### 10.3. Beneficial Effects on Diseases

The beneficial effects of kefiran in different animal models regarding several diseases have also been reported. Specifically, kefiran suppressed the blood pressure and reduced lipid concentration in serum and liver of hypertensive rats, lowered blood glucose in genetically diabetic mice, and alleviated constipation symptoms in rats fed with a low-fiber diet [151]. Furthermore, kefiran accelerated sterol excretion, protected hepatic injuries and decreased histamine excretion in cecum content and feces in two rat models [152]. The therapeutic potential of kefiran regarding allergic bronchial asthma has also been indicated in a murine model as reduced inflammation of lung tissue and airway hyper-responsiveness were determined in the tested animals receiving the polysaccharide [153]. Kefiran has been shown to reduce atherosclerosis in rabbits fed a high-fat diet and this effect was attributed to its antioxidant and anti-inflammatory action [154].

### 10.4. Antitumor Activity

Back in 1982, the antitumor activity of a water-soluble polysaccharide isolated from kefir grains was studied in mice subcutaneously inoculated with Ehrlich carcinoma or Sarcoma 180, and it was found that the tumor growth was inhibited either by oral or by intraperitoneal administration [155]. Kefiran also exhibited anticancer properties when two human carcinoma cell lines (HeLa and HepG2) were used as it was found to affect their viability in a dose-dependent manner by also changing their morphological characteristics [156]. In the same study, kefiran’s effects on the mortality and abnormality development of zebrafish embryos were investigated and the authors concluded that kefiran should be considered a potential anticancer agent without toxic side-effects.

### 10.5. Gut Microbiota Modulation

The alteration of gut microbiota and its important role on gut homeostasis has also been reported. In this context, kefiran was able to promote the growth of *Bifidobacterium bifidum* PRL2010 and to modulate its gene expression regarding metabolism of dietary glycans and host–microbe effector molecules, thus increasing its potential as a probiotic therapy agent [102]. Moreover, kefiran administered ad libitum in the drinking water of mice enhanced *Bifidobacterium* populations while no changes were observed in *Lactobacillus* ones, suggesting kefiran as a bifidogenic functional ingredient [157].

### 10.6. Other Properties

Nematodes have also been used as model organisms that were fed kefiran produced from rice fermented with *L. kefiranofaciens*, along with *E. coli* OP50 [158]. It was shown that not only the lifespan of nematodes was extended, but also anti-aging and heat stress tolerance effects were observed.

Last but not least, the investigation of biopolymers, especially kefiran, in order to make three-dimensional porous scaffolds used for tissue engineering revealed that polymeric kefiran scaffolds, which are characterized by high porosity structure and controllable morphology, are promising matrices in terms of biomedical applications [106,159].

## 11. Use in Dairy Products

### 11.1. Production of Kefir Grains and Beverages Containing L. kefiranofaciens

Consumption of kefir has increased worldwide, and by 2023, the market is expected to reach 1.85 billion $US [160]. *L. kefiranofaciens* is undoubtedly among the most important species and has therefore been chosen for kefir production. The species dominates kefir grains [29,34,35,46,51] and influences grain formation, the growth of other microbial species in the consortium and the organoleptic characteristics of the final product [34,35].

Kefir beverages can be produced by fermenting cow, goat, buffalo, sheep, camel, mare, and donkey milk with kefir grains according to the “Russian method”, which is essentially a “back-slopping” procedure and can be repeated ad infinitum [1,30,161,162]. In Russia, kefir has also been produced by inoculating pasteurized milk with kefir beverage, a mother culture prepared by carrying out traditional kefir fermentation and sieving the grains [163]. Moreover, a new technique to produce kefir using immobilized starter cultures isolated from kefir grains, among them *L. kefiranofaciens*, has been developed [164]. In this case, various LAB and yeast strains, entrapped in microspheres, are used for kefir beverage production, which is microbiologically similar to the original kefir beverage.

However, nowadays, kefir grains are also industrially produced and commercialized by various companies worldwide and it seems that *L. kefiranofaciens* is one of the key microbial species of the blends as well. Species-level classifications via 16S rRNA gene sequencing and shotgun metagenomic sequencing identified six dominant bacterial species of *Lactobacillus* in a commercial kefir grain sample (Fusion Teas, McKinney, TX, USA) with *L. kefiranofaciens* being the predominant species among them [56]. *L. kefiranofaciens* has also been detected in the commercial kefir grains of a biotechnology company (Bionova snc, Villanova sull’Arda, Piacenza, Italy) [23]. Nejati et al. [38] reported the detection of *L. kefiranofaciens* in two commercial kefir grains and their respective beverages (Primal Life UG, Berlin, Germany). Moreover, Wang et al. [65] studied commercial grains purchased from Huacheng Biological Corporation (Changchun, Jilin, China), Mr. Pro Company (Germany) and Mr. and Mrs. Kefir Company (USA), and their taxonomic analysis showed *L. kefiranofaciens* being among the main species in all grains despite their different origins.

When LAB, including *L. kefiranofaciens*, and yeasts are used as starters for the industrial production of kefir beverages, it is difficult to sustain the necessary stable and constant consortium needed for manufacturing a standardized quality final product, because of the complex microbiological composition of the grains [164]. However, it has been shown that when grains are used, the kefir beverage is more desirable in comparison to kefir produced with a starter culture containing a less rich cocktail of strains [165].

### 11.2. Cheese Production Containing L. kefiranofaciens

Interestingly, there is currently a trend to use kefir grains or kefir beverages as starter cultures in cheese production to exploit their potential effect on the quality, health, and safety properties of the final product [166]. Various cheese types have been produced this way, such as a hard-type cheese using thermally-dried free and immobilized kefir cells [167], as well as Feta-type and whey-cheeses using freeze-dried cultures isolated from commercially available kefir grains [168]. However, in both cases the cheese microbiota has not been analyzed, so the occurrence of *L. kefiranofaciens* cannot be verified.

On the other hand, when the EPS producing *L. kefiranofaciens* ZW3 strain, isolated from Tibetan kefir grains [21], has been used as an adjunct culture in Mozzarella-type cheese production from cow milk, along with strains of *L. bulgaricus* and *S. thermophilus* [169], a balanced symbiosis with the other LAB strains used was observed while *E. coli* and fungal populations were detected at low levels during cheese ripening. Additionally, a freeze-dried Tibetan kefir co-culture containing *L. kefiranofaciens* was used as a starter culture in Camembert-type cheese production for the first time [80]. SEM analysis showed that the microbiota was dominated by a variety of lactobacilli in close association with yeasts, while *L. kefiranofaciens* was identified by PCR-DGGE and isolated during ripening.

### 11.3. Other Products

A cheese whey-based fermented beverage has also been produced using kefir grains as starter culture, as nowadays novel trends in whey exploitation are emerging [47]. An analysis of the beverage revealed a steady structure and dominant microbiota including *L. kefiranofaciens* both subsp. *kefiranofaciens* and subsp. *kefirgranum*.

*L. kefiranofaciens* was also used in rice fermentation in order to produce kefiran [158] and finally a novel immobilized system using kefir LAB, among them *L. kefiranofaciens*, and sugar cane pieces was developed to produce fermented milk [170].

## 12. Conclusions

The vast majority of the kefir-derived lactobacilli belong to the *L. kefiranofaciens* species, which are Gram-positive, nonmotile, capsulated, non-spore-forming rods and includes two subspecies, namely *kefiranofaciens* and *kefirgranum*. The increased research interest in the purported health-related benefits of kefir itself has led to extended research of the microorganisms of highest abundance in this complex ecological niche. Among them *L. kefiranofaciens* is outstanding and has been revealed to exhibit various probiotic properties, such as antimicrobial activity, immunomodulation, reduction in cholesterol levels, risk of allergies and cancer. Some of them are mainly associated with the water-soluble EPS kefiran, which is produced by the subspecies *kefiranofaciens* and is famous for its technological properties and its potential to be used as a stabilizer, emulsifier, thickener, gelling agent and fat substitute, as well as its multiple health benefits. Taking into consideration that *L. kefiranofaciens* is isolated from the natural microbial consortium of kefir, it is comprehensible why it is a fastidious microorganism demanding complex growth media, anaerobic conditions and more than one day to grow. Its growth and beneficial properties in the natural environment of a fermented food undoubtedly depend on its coculture with other microorganisms, among them *S. cerevisiae*. However, as difficult as it might be, it is worth making use of it in the food or pharmaceutical industry to take advantage of its substantial benefits.

## Figures and Tables

**Figure 1 microorganisms-09-02158-f001:**
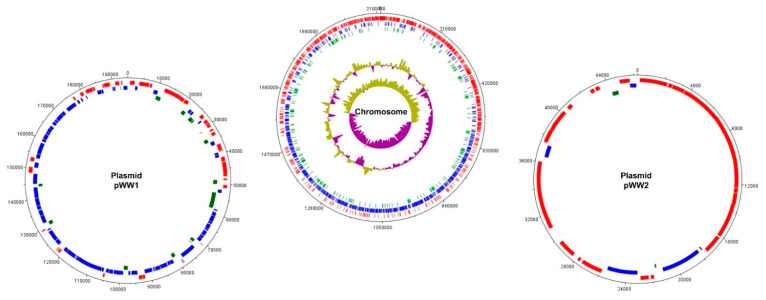
Circular maps of the *L. kefiranofaciens* ZW3 chromosome and plasmids pWW1 and pWW2 generated by DNAPlotter [14]. Labeling from the outside to the inside of the circle, each ring contains information on the chromosome: CDSs on the forward strand (red); CDSs on the reverse strand (blue); Putative pseudogenes (green); GC content; GC skew.

**Figure 2 microorganisms-09-02158-f002:**
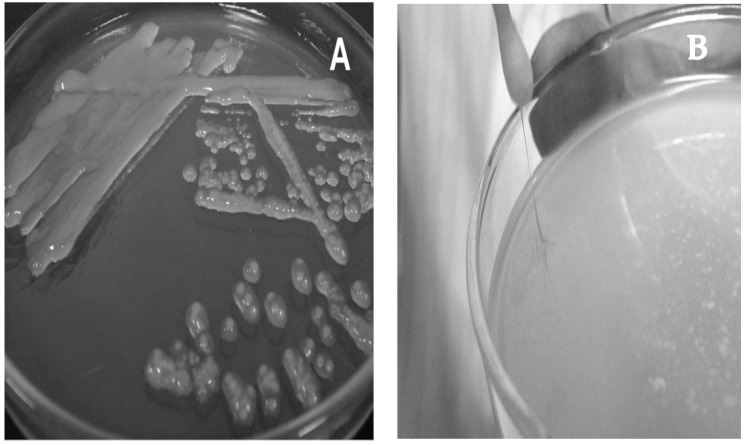
(**A**) Slimy colonies of *L. kefiranofaciens* 1P3 in medium containing sucrose [33]. (**B**) Ropy behavior of colonies of *L. kefiranofaciens* ZW3 [21].

**Figure 3 microorganisms-09-02158-f003:**
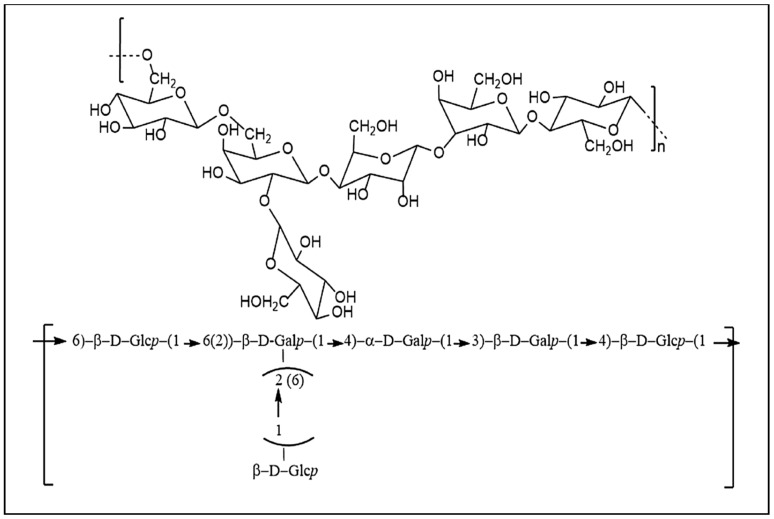
Proposed molecular structure of the heteropolysaccharide kefiran—chair configuration [101].

**Figure 4 microorganisms-09-02158-f004:**
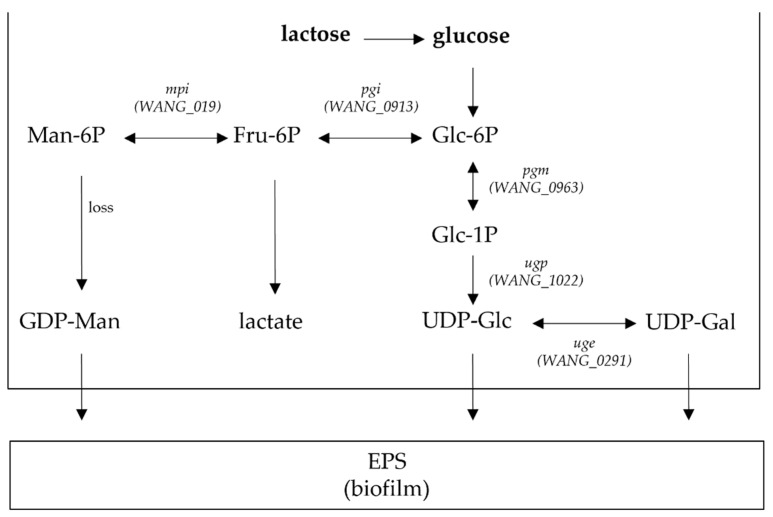
The proposed biosynthesis pathway for EPS production according to KEGG analysis [104]. *mpi*: mannose-6-phosphate isomerase; *pgi*: glucose-6-phosphate isomerase; *pgm*: α-phosphoglucose mutase; *ugp*: UDP-glucose pyrophosphorylase; *uge*: UDP-galactose 4-epimerase.

**Table 1 microorganisms-09-02158-t001:** Sequences of the species-specific primers and probes targeting *Lactobacillus kefiranofaciens* as applied in chronological order.

PCR Type	Primer/Probe	Sequence (5′-3′)	Reference
Species-specific PCR	Forward (R2C2-16SF)	TAAGAAAGCA GTTCGCATGA ACAG	[20] Patent US 2009/0130.072 A1
	Reverse (R2C2-16SR)	GGGACTTTGT ATCTCTACAA ATGG	
Real-time PCR	Forward	CAGTTCGCATGAACAGCTTTTAA	[37]
	Reverse	GCACCGCGGGTCCAT	
	Probe	FAM-CGCAAGCTGTCGCTAA-TAMRA	
Colony species-specific PCR, qPCR	Forward (LK1-2F)	GAGCGGAACCAGCAGAATCA	[28]
	Reverse (LK1-2R)	GCTGTTCATGCGAACTGCTT	
	Probe	FITH-CCACCGCTACACATGGAGTTCTAC	
Multiplexed qPCR	Forward	GCAACAACCAAAGTATTGTA	[38]
	Reverse	TAGCCGAAGAGGATCTAA	
	Probe	Q705-ACC[+A]CA[+T]CA[+C]CA[+A]CTCTAA-BHQ3	

**Table 2 microorganisms-09-02158-t002:** *Lactobacillus kefiranofaciens* detection in and isolation from kefir grains and beverages in chronological order.

Origin	Detection	Number of Isolates	Reference
Grains, Denmark	CD	Ropy colonies	[12]
Grains, Denmark	CD	4 of the above colonies	[3]
Grains, Denmark	CD	26	[39]
Grains, France, Denmark, Canada	CD	14 previously isolated	[10]
Grains, Russia	CD	3	[18]
Grains, Belgium	CI	-	[49]
Grains, Taiwan	CD, CI	3	[26]
Grains, China	CD	Ropy colonies	[21]
Grains, China	CI	-	[45]
Grains and beverages, Brazil	CI	-	[47]
Grains and beverages, Ireland	CI	-	[31]
Grains and beverages, Turkey	CD, CI	Not detected	[29]
Grains, Turkey	CI	-	[55]
Grains, Brazil	CI	-	[46]
Grains, Slovenia	CD, CI	40	[48]
Grains, Argentina	CD, CI	11	[27]
Grains, Turkey	CI	-	[51]
Grains, China	CD, CI	11	[44]
Grains, Italy	CD, CI	-	[23]
Grains, Korea	CI	-	[37]
Grains, Belgium	CI	-	[50]
Grains, Brazil	CD, CI	5	[22]
Grains, USA	CI	-	[56]
Grains, Russia	CD, CI	Not detected	[30]
Grains and beverages, Ireland, France, United Kingdom	CI	-	[34]
Grains, Malaysia	CI	-	[57]
Grains, Turkey	CI	-	[35]
Grains, China	CD, CI	18	[28]
Kefir and beverages, China	CI	-	[58]
Beverages, Korea	CI	-	[59]
Grains, China	CI	-	[60]
Grains, Turkey	CD	3	[61]
Grains, Greece	CI	-	[62]
Beverage, Korea	CI	-	[63]
Grains and beverages, Germany, Italy	CI	-	[38]
Grains and beverages, United Kingdom, Caucuses region, Ireland, Lithuania, South Korea, USA	CI	-	[64]
Grains, China, Germany, USA	CI	-	[65]
Grains, China	CD, CI	1	[66]

CD: culture-dependent method; CI: culture-independent method; -: isolation not performed.

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
