# Peer review of "Lactobacillus kefiranofaciens: From Isolation and Taxonomy to Probiotic Properties and Applications"

_microorganisms, 2021, doi:10.3390/microorganisms9102158_

Round 1
Reviewer 1 Report
The paper addresses a very interesting topic related to characterization as well describing technological as well as functional properties of Lactobacillus kefiranofaciens including health-promoting characteristics of kefiran produced by this microorganism. The manuscript discusses aspects related to the potential for biomedical applications and the possibility of using the bacteria as starter cultures in food technology. I assess the work as well prepared, only below are indicated minor remarks for verification and possible correction of the text.
- Although the title is quite intriguing, I suggest modifying it to emphasize the more scientific nature of the paper.
- Lactobacillus parakefir or Lactobacillus parakefiri?- check in whole text
- Abstract Lines 12-15: This fragment of text could be improved for better understanding the presented information (precise if "it" refers to kefir grains)
- Lines 70-73: the indicated fragment of text needs to be modified to express the included information in a more consistently and clearly way
Author Response
First, we would like to thank the Reviewer 1 for the positive feedback, as well as for the comments/suggestions, which helped us to improve our manuscript. Please find below our point-by-point replies.
- Although the title is quite intriguing, I suggest modifying it to emphasize the more scientific nature of the paper.
Reply: The title has been modified to “Lactobacillus kefiranofaciens: from isolation and taxonomy to probiotic properties and applications”.
- Lactobacillus parakefir or Lactobacillus parakefiri?- check in whole text.
Reply: The correct species name (basonym) is Lactobacillus parakefir. It has been checked throughout the text (line 33), while the new species name is Lentilactobacillus parakefiri (line 32).
- Abstract Lines 12-15: This fragment of text could be improved for better understanding the presented information (precise if "it" refers to kefir grains).
Reply: “It” refers to L. kefiranofaciens. It has been replaced so that information is now clearly presented (line 13).
- Lines 70-73: the indicated fragment of text needs to be modified to express the included information in a more consistently and clearly way.
Reply: The text in lines 70-73 has been modified so that information is now clearly presented (now lines 71-77).
Reviewer 2 Report
This review article summarizes the origin, taxanomy, strain characteristic, functionality and food application of kefir isolated bacteria, Lactobacillus kefiranofaciens. The author provides a comprehensive information on this strain and addresses its important role for food and pharmaceutical industry. However, this whole manuscript needs to reorganize and remove the redundant parts.
Comments:
Major
- Please change the title. Magic is not a scientific term.
- L306: The information in Table 2 is not necessary in this paper. The isolating and identifying approaches including culture-dependent and culture-independent conditions have been clearly described in section 3-5. Remove or shorten it.
- Subtitle could assist the readers to catch the key point. Several sections need to provide subtitles including 5. Species and subspecies discrimination, 7. EPS, and 9. Functional and probiotic properties.
- L895: 11. Use in dairy products: The most of the products mention here is kefir. Lactobacillus kefiranofaciens is original isolated from Kefir grain, which is not the application. Using kefir as starter culture for cheese making is not an application for Lactobacillus kefiranofaciens. Please rewrite this section.
Minor:
Line 31: “Lentilactobacillus kefiri” should be “Lentilactobacillus parakefiri” (basonym: Lactobacillus parakefiri)?
Line 457-459: Please remove this sentence “Use of Tibeten kefir as starter culture in Camembert cheese making for the first time”, which has been mentioned in line 951-952.
Line 922-931: Some phrases are redundant.
Author Response
First, we would like to thank the Reviewer 2 for the positive feedback, as well as for the comments/suggestions, which helped us to improve our manuscript. Please find below our point-by-point replies.
Major
- Please change the title. Magic is not a scientific term.
Reply: The title has been modified to “Lactobacillus kefiranofaciens: from isolation and taxonomy to probiotic properties and applications”.
- L306: The information in Table 2 is not necessary in this paper. The isolating and identifying approaches including culture-dependent and culture-independent conditions have been clearly described in section 3-5. Remove or shorten it.
Reply: Table 2 has not been removed, as it contains several references not mentioned elsewhere in the text. It has been significantly shortened, therefore information regarding isolation and identification including culture-dependent and culture-independent approaches has been removed.
- Subtitle could assist the readers to catch the key point. Several sections need to provide subtitles including 5. Species and subspecies discrimination, 7. EPS, and 9. Functional and probiotic properties.
Reply: Subtitles have been added in sections 5, 7, 9 and 10.
- L895: 11. Use in dairy products: The most of the products mention here is kefir. Lactobacillus kefiranofaciensis original isolated from Kefir grain, which is not the application. Using kefir as starter culture for cheese making is not an application for Lactobacillus kefiranofaciens. Please rewrite this section.
Reply: Although L. kefiranofaciens is originally isolated from kefir grains, the facts that 1) kefir grains and beverages containing L. kefiranofaciens are industrially produced and commercialized by various companies and 2) kefir grains or kefir beverages can be further used (in terms of containing starter cultures, including L. kefiranofaciens strains) for the production of various dairy products, can be considered as an application of the species as well.
In this concept and after taking into account the reviewer’s comment, three subtitles have been added in this section to clarify the use of L. kefiranofaciens in dairy products and the text has been revised.
In the first paragraph entitled “11.1. Production of kefir grains and beverages containing L. kefiranofaciens” general information is first given about kefir production and then information is provided about the increasing willing of the food industry to produce kefir products containing the specific species.
Respectively, in the second paragraph entitled “11.2. Cheese production containing L. kefiranofaciens” introductory information is given about the use of kefir in cheese production to further report the specific use of the species L. kefiranofaciens as an adjunct culture.
Minor:
Line 31: “Lentilactobacillus kefiri” should be “Lentilactobacillus parakefiri” (basonym: Lactobacillus parakefiri)?
Reply: The species name has been corrected (lines 32-33). The correct species name is Lentilactobacillus parakefiri, but the basonym was Lactobacillus parakefir (after Takizawa et al. 1994).
Line 457-459: Please remove this sentence “Use of Tibeten kefir as starter culture in Camembert cheese making for the first time”, which has been mentioned in line 951-952.
Reply: This sentence has now been removed (lines 451-453).
Line 922-931: Some phrases are redundant.
Reply: In this paragraph some phrases have now been removed (lines 946-957).